# Unified Low-Resource Sequence Labeling by Sample-Aware Dynamic Sparse Finetuning

**Sarkar Snigdha Sarathi Das[1], Ranran Haoran Zhang[1], Peng Shi[2], Wenpeng Yin[1], Rui Zhang[1]**

[1]Pennsylvania State University, [2]University of Waterloo

`{sfd5525, haoranz6, wenpeng, rmz5227}@psu.edu;`
`peng.shi@uwaterloo.ca`

## Abstract

Unified Sequence Labeling articulates different sequence labeling tasks such as Named Entity Recognition, Relation Extraction, Semantic Role Labeling, etc. in a generalized sequence-to-sequence format. Unfortunately, this requires formatting different tasks into specialized augmented formats which are unfamiliar to the base pretrained language model (PLMs). This necessitates model fine-tuning and significantly bounds its usefulness in data-limited settings where fine-tuning large models cannot properly generalize to the target format. To address this challenge and leverage PLM knowledge effectively, we propose FISH-DIP, a sample-aware dynamic sparse finetuning strategy. It selectively finetunes a fraction of parameters informed by highly regressing examples during the fine-tuning process. By leveraging the dynamism of sparsity, our approach mitigates the impact of well-learned samples and prioritizes underperforming instances for improvement in generalization.

Across five tasks of sequence labeling, we demonstrate that FISH-DIP can smoothly optimize the model in low-resource settings, offering up to 40% performance improvements over full fine-tuning depending on target evaluation settings. Also, compared to in-context learning and other parameter-efficient fine-tuning (PEFT) approaches, FISH-DIP performs comparably or better, notably in extreme low-resource settings. The source code of FISH-DIP will be available at: https://github.com/psunlpgroup/FISH-DIP

## 1 Introduction

Sequence Labeling tasks such as Named Entity Recognition, Relation Extraction, Semantic Role Labeling, etc. aim to assign target labels to the components of a sequence from a set of categorical classes. Accurate sequence labeling requires learning complex structural dependency, where even state-of-the-art pretrained LLMs like GPT-3.5-turbo fail to show acceptable performance (Li et al.,

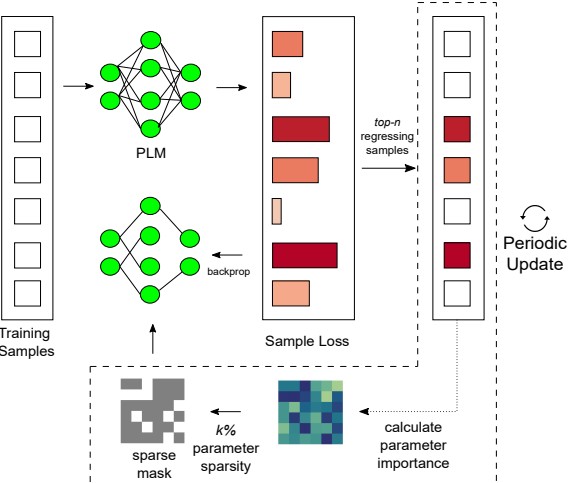

Figure 1: Overview of FISH-DIP for Unified Low-Resource Sequence Labeling. All augmented training samples (Table 1) are first channeled through PLMs (e.g. T5-large) to calculate the training loss. Top-$n$ most regressing samples are chosen for calculating parameter importance, for k% sparsity mask. In backpropagation, this sparsity mask is applied for sparse finetuning. Parameter importance is updated dynamically using the feedback from underperforming samples.

2023). A reasonable way to address this gap is to linearize these structured prediction tasks into an augmented lanuage representation (Paolini et al., 2021; Liu et al., 2022b) that frames all sequence labeling tasks in a unified sequence-to-sequence format (See examples in Table 1). This allows us to capture the structured information while aligning the prediction and training objectives of large language models. Significantly, when paired with this unified format, the pretrained language models (PLMs) exhibit an impressive ability to effectively generalize to different tasks, capitalizing on the inherent potential for knowledge transfer.

Despite these advantages, fine-tuning PLMs to accommodate such an "augmented" format for sequence labeling in data-limited low-resource environments is significantly challenging, which has

| Task $\mathcal{T}$ | Input $X$ | Transformation $\mathcal{F}_{\mathcal{T}}$ that produces Augmented NL Target $\bar{X}$ |
|---|---|---|
| NER | Over 900 million US dollars of foreign capital was actually utilized, increasing nearly 40% compared with the same period previous year. | [ Over 900 million US dollars | *monetary* ] of foreign capital was actually utilized , increasing [ nearly 40 % | *percent* ] compared with the same period [ previous year | *date* ] . |
| Relation Extraction | In the Middle Adriatic Basin, there is evidence of Permian volcanism observed on the Vis island and as volcanic islands of Jabuka and Brusnik. | relationship between [ Jabuka ] and [ Adriatic Basin ] = *located in or next to body of water* |
| Joint ER extraction | In 1831 , the 20th President of the United States, James Garfield, was born in Orange , Ohio. | In 1831 , the 20th President of the [ United States | *location* ] , [ James Garfield | *person* | *lives in = Orange, Ohio* ] , was born in [ Orange , Ohio | *location* ] . |
| DST | [User]: hi, can you help me find a place to stay on the north side?; [Agent]: I have 13 hotels on the north side of town, do you have any preferences [User]: yes, are there any expensive ones? I also would like free parking as well. | [ belief ] *hotel area* north, *hotel book day* not given, *hotel book people* not given, *hotel book stay* not given, *hotel internet* not given, *hotel name* not given, *hotel parking* yes, *hotel price range* expensive, *hotel stars* not given, *hotel type* hotel [ belief ] |
| SRL | The lawyers have renewed their arguments in Texas and eight other states where the defense is [ permitted ] under state law. | The lawyers have renewed their arguments in [ Texas and eight other states | *AM-LOC* ] [ where | *R-AM-LOC* ] [ the defense | *A1* ] is permitted [ under state law | *AM-LOC* ]. |

Table 1: We employ augmented language to unified sequence labeling tasks. Example transformations for Named Entity Recognition (NER), Relation Extraction, Joint Entity Relation extraction, Dialogue State Tracking (DST), and Semantic Role Labeling (SRL).

not been thoroughly investigated previously. For instance, most state-of-the-art Parameter Efficient Fine-Tuning (PEFT) methods, such as LoRA (Hu et al., 2021), Adapter (Houlsby et al., 2019), Compactor (Karimi Mahabadi et al., 2021), $(IA)^3$ (Liu et al., 2022a), etc., introduce new trainable parameters for altering the output. Although effective in sentence classification or generation tasks, we find these methods often underperform in data-limited low-resource sequence labeling tasks as shown in our results, primarily due to the insufficient available data for training the added parameters to accommodate the new augmented language format.

In the context of data-limited low-resource universal sequence labeling tasks, we identified fixed Fisher Information sparsity training (Sung et al., 2021) as a promising direction. Instead of introducing new parameters to train from scratch, it selects and finetunes the most influential parameters from the existing well-trained PLMs, yielding results on par with full fine-tuning (fine-tuning all the parameters of the model). However, our investigations reveal that a sample-aware dynamic selection of parameter importance can lead to a significant performance boost in few-sample settings. Thus, we propose leveraging **FISH**er information for Sparse Fine-tuning with **D**ynamic **I**mportance of **P**arameters (**FISH-DIP**), a method of sparse fine-tuning that dynamically chooses and adjusts a small fraction of the most relevant parameters based on the feedback from the available samples with the highest loss, prioritizing those that exhibit highest deviation from the desired outcome. As shown in Figure 1, augmented natural language training sam-

ples are channeled through PLMs (e.g. T5-large), from which we can calculate losses for each of the samples. Then under-performing samples are chosen for calculating parameter importance, which in turn is leveraged to determine a sparsity mask for sparse finetuning. As training progresses, the importance of parameters is updated to accommodate the evolving learning requirements of both the model parameters and training samples. Consequently, FISH-DIP ensures that the fraction of the model parameters chosen is primarily influenced by the most pertinent samples, effectively helping in smoother optimization (Figure 4) and generalization while sustaining the enhancement in few-shot performance.

We conduct comprehensive testing of FISH-DIP across a range of tasks, including Named Entity Recognition (Sang and De Meulder, 2003; Weischedel et al., 2013), Relation Extraction (Han et al., 2018), Joint Entity-Relation Extraction (Walker et al., 2006), Semantic Role Labeling (Carreras and Màrquez, 2005), and Dialogue State Tracking (Eric et al., 2019). Our findings reveal that across diverse few-shot settings, FISH-DIP significantly enhances the few-shot performance of universal sequence labeling tasks. Besides, unlike other methods (Lu et al., 2022) that require large-scale data and pretraining, FISH-DIP only relies on prudent optimization strategy to augment performance. This unique feature makes it particularly valuable for future applications in low-resource language contexts, where external resources may be extremely scarce. Furthermore, FISH-DIP consistently outperforms other popular Parameter Efficient Finetuning (PEFT) methods, demonstrating

the efficacy of dynamic parameter selection in low resource settings. Our contributions are as below:

- We broadly explore the field of *data-limited low-resource sequence labeling from a unified perspective*, a domain that has predominantly remained underexplored in spite of the inherent challenges it presents in capturing structural information.

- We propose a sample-aware dynamic sparse fine-tuning scheme, FISH-DIP, that dynamically updates the parameter importance with the feedback from the most regressing samples, subsiding the influence of already trained samples. This helps in better generalization, particularly in low-resource settings.

- We rigorously test our proposed method across five tasks in seven datasets in a wide range of low-resource settings. In most cases, FISH-DIP outperforms other approaches while finetuning only a fraction of the parameters of the model.

## 2  FISH-DIP for Unified Low-Resource Sequence Labeling

In this section, we first discuss how different sequence labeling problems can be transformed into a unified "augmented" format (Section 2.1). Next, we discuss how Fisher information is leveraged to calculate parameter importance (Section 2.2). Finally, we discuss how FISH-DIP leverages sample-aware dynamism to boost few-sample performance in sequence labeling (Section 2.3).

### 2.1  Unified Sequence Labeling by Augmented Language

Given a sequence labeling task $\mathcal{T}$, and few-shot training samples $(x_i, y_i) \in \mathcal{T}$ where $x_i$ denotes the input sentences, and $y_i$ denotes the associated ground truth predictions (e.g., target entities, relations, etc.), we can apply transformation function $\mathcal{F}_{\mathcal{T}}$ such that $\mathcal{F}_{\mathcal{T}}(x_i, y_i, S)$ results in augmented language text $\bar{x}_i$ using special tokens $S$ where $S = \{[,],|,=\}$. This essentially converts standard sequence prediction tasks into sequence-to-sequence format, and the output text can encode all the structured information required for sequence labeling. Some examples of such transformation $\mathcal{F}_{\mathcal{T}}$ used in Paolini et al. (2021) are given in Table 1.

After a well-trained model generates an output sequence, specialized decoding schemes are utilized to properly parse the output. Specifically, all special tokens are removed, and class-type information is extracted to produce a cleaned output. Then, at the token level, Dynamic Programming-based Needleman-Wunsch alignment algorithm (Needleman and Wunsch, 1970) is used for robust alignment of inputs and outputs, even in the case of noisy generation. Output classes that do not match any target types are discarded in the process.

### 2.2  Fisher Information and Sparsity Mask

Fisher information indicates the importance of a parameter for a particular task. Prior works in machine learning have leveraged this for pruning and model compression (Theis et al., 2018; Singh and Alistarh, 2020). In this regard, Fisher information matrix $F_\theta \in \mathbb{R}^{|\theta| \times |\theta|}$ needs to be calculated. However, with pretrained language models, it is computationally intractable because of the $|\theta| \times |\theta|$ scale of computation where $\theta$ is the number of parameters. To get around this computational complexity, Sung et al. (2021) proposed using average of the squared gradient of the model's output as approximate Fisher information. Although it requires taking expectation over ouput distribution, in supervised setting due to the availability of label $y_i$, this "Empirical Fisher" can be simply calculated for "heuristic" paramter importance as below:

$$\hat{F}_\theta \approx \frac{1}{N} \sum_{i=1}^{N} (\nabla_\theta \log p_\theta(y_i|x_i))^2 \qquad (1)$$

where $p_\theta(y_i|x_i)$ denotes the output probability $y_i$ with parameters $\theta$ given input $x_i$, $N$ is the number of samples, and $\hat{F}_\theta \in \mathbb{R}^{|\theta|}$. Using the calculated empirical Fisher information of the parameters, top $k\%$ parameters can be kept for creating a FISH mask for sparse fine-tuning, while keeping the rest of the network fixed. Sung et al. (2021) found that calculating this FISH mask at the start of training with a small number of randomly selected training samples (e.g. $N \approx 1024$) and keeping the importance mask *fixed* during training can provide a similar performance as full finetuning.

### 2.3  Dynamic Sample Aware Sparse Tuning : FISH-DIP

In low-resource settings, employing fixed sparsity can quickly fit the limited number of training samples. However, this accelerated learning process

can result in a rapid change of parameter importance, thereby necessitating the update of the parameter importance. With fixed sparsity mask, decreasing losses for some samples may cause increased losses for others, resulting in an overall irregular update trend.

To alleviate these issues, we propose dynamic sample aware sparse fine-tuning, where we periodically update parameter importance based on the feedback from most regressing training samples. Since samples with lower training errors do not take part in parameter selection, their associated parameters remain unperturbed. This approach helps mitigate overfitting and fosters a smoother training progression (Figure 4). More specifically,

$$\hat{F}_\theta \approx \frac{1}{n} \sum_{\substack{\{(x_i,y_i)|\mathcal{L}_{tr}(x_i,y_i) \\ \in \text{top}_n\}}} (\nabla_\theta \log p_\theta(y_i|x_i))^2 \quad (2)$$

where $\text{top}_n$ indicates the losses of top-$n$ most regressing training examples. Algorithm 1 demonstrates how FISH-DIP can be used in finetuning the model in low-resource sequence labeling.

In short, this dynamic calculation of parameter importance helps us in two fronts: (i) We recalculate the parameter importance so that new parameters that are chosen for training focus on regressing examples. Similar to boosting algorithms, they try to optimize the less performant examples to ensure overall smooth optimization across samples (Figure 4). (ii) Samples that have already been optimized do not take part in feedback process for parameter importance calculation. As a result, they cannot overfit the model resulting in improved low-resource performance from finetuning.

## 3 Experiment Setups

We describe our tasks and datasets, evaluation settings, and different types of baselines.

**Tasks and Datsets** To evaluate the efficacy of FISH-DIP, we cover five different tasks across seven datasets in Named Entity Recognition (Sang and De Meulder, 2003; Weischedel et al., 2013), Relation Extraction (Han et al., 2018), Joint Entity Relation Extraction (Roth and Yih, 2004; Walker et al., 2006), Dialogue State Tracking (Eric et al., 2019), and Semantic Role Labeling (Carreras and Màrquez, 2005) tasks. For evaluation of NER and SRL, we use entity F1-score, while in joint ER we use entity and relation F1 scores. Finally, in DST,

---

**Algorithm 1:** FISH-DIP

**Require:** Low-resource training Data $X_{tr}$, Train loss $L_{tr}$, Augmentation function $\mathcal{F}_\mathcal{T}$, special tokens $S$, Pretrained Language Model PLM, sparsity $k\%$, update steps $m$, regressing examples $n$, total training steps $T$

Initialize sparse mask $M \in \mathbb{R}^{|\theta|}$ to zeros

**for** $t$ *in range(0, T)* **do**

  Sample minibatch $(X, Y) \in X_{tr}$
  $\bar{X} = \mathcal{F}_\mathcal{T}(X, Y, S)$ //e.g., Table 1
  $X' = PLM(X)$
  $losses = L_{tr}(X', \bar{X})$
  **if** $t \mod m = 0$ **then**

    // Recalculate parameter importance and sparsity mask
    Calculate parameter importance $\hat{F}_\theta$ using Eq. 2 with $top_n$ most regressing examples
    *sort* all the parameters using $\hat{F}_\theta$ and select top $k\%$ parameters
    Reset $M$ and set the selected parameters to 1

  **end**
  Backprop and Apply sparsity mask $M$
  Update Parameters

**end**

---

we use joint accuracy as evaluation metric. Table 2 summarizes different datasets and evaluation metrics used for each of these tasks.

**Low-resource Settings** We compare the performance of FISH-DIP across different tasks and datasets in a wide range of few-sample settings. In CoNLL'03, CoNLL'04, and OntoNotes we evaluate the performance with 1%, 5%, and 10% of the full training dataset. For large datasets e.g. CoNLL'05 SRL and MultiWoz 2.1 dataset, we compare the performance in 0.5%, 1%, and 5% data settings. Finally, in FewRel, we evaluate the performance in 5-way 1-shot, 5-way 5-shot, 10-way 1-shot, and 10-way 5-shot settings as standard.

**Baselines** We compare the efficacy of FISH-DIP against (i) full finetuning based baselines that include task-specific state-of-the-art methods, TANL (Paolini et al., 2021), and constrained generation-based method ASP (T5-large) (Liu et al., 2022b) (ii) sparse Finetuning baseline FISH mask (Sung

| Task | Dataset | Low-resource Setting | Metric |
|------|---------|---------------------|--------|
| NER | CoNLL'03, OntoNotes | 1%, 5%, 10% | F1 |
| Relation Extraction | FewRel 1.0 | 5/10 way, 1/5 shot | F1 |
| Joint ER | CoNLL '04, ACE2005 | 1%, 5%, 10% | Entity F1, Relation F1 |
| Dialogue State Tracking | MultiWoz 2.1 | 0.5%, 1%, 5% | Joint Accuracy |
| Semantic Role Labeling | CoNLL'05 SRL WSJ and Brown | 0.5%, 1%, 5% | F1 |

Table 2: Our experiments cover diverse and comprehensive sequence labeling tasks, datasets, and evaluation settings.

et al., 2021) in 1% and 5% sparsity, and (iii) in-context baseline when applicable. For all methods, we use T5-large as the backbone PLM. Table 3-7 lists these comparison results in detail across different tasks, datasets, and settings. We discuss these tasks, datasets, and baselines in more detail in Appendix B. Finally, we also test the efficacy of FISH-DIP against different parameter-efficient finetuning approaches, such as Adapter (Houlsby et al., 2019), Compacter (Karimi Mahabadi et al., 2021), LoRA (Hu et al., 2021), $(IA)^3$ (Liu et al., 2022a), and prefix tuning (Li and Liang, 2021) as shown in Figure 2.

## 4 Results and Analysis

Our results and analysis aim to answer the following research questions:

- RQ 1: What is the overall efficacy of FISH-DIP across low-resource sequence labeling tasks compared with baselines (Section 4.1)?
- RQ 2: How does FISH-DIP perform when compared with other PEFT approaches (Section 4.2)?
- RQ 3: How does FISH-DIP compare with in-context learning using LLMs (Section 4.3)?
- RQ 4: What is the effect of dynamically choosing regressing samples for calculating parameter importance for sparsity mask (Section 4.4)?
- RQ 5: How is performance of FISH-DIP influenced by the percentage of sparsity (Section 4.5)?

### 4.1 Overall Results

Table 3 - 7 demonstrates that across different low-resource settings, FISH-DIP finetuning offers substantial performance improvement over all other baselines. In particular, for very low resource settings (e.g. 1% training data) FISH-DIP does not only offer significant performance uplift but also shows lower performance variance demonstrating its robustness. We also find that, while trained in identical settings, fixed sparse masks (1%/5% spar-

sity) (Sung et al., 2021) often loses performance against full finetuning counterpart. This particularly demonstrates how importance of parameters changes over the course of training - which can be extremely useful in improving performance in low-resource sequence labeling tasks. In named entity recognition (CoNLL'03 and OntoNotes dataset), FISH-DIP shows improved performance over all the other baselines while updating only a fraction of the parameters. On the other hand, in dedicated few-shot relation extraction dataset FewRel that uses traditional N-way K-shot settings, FISH-DIP shows substantial performance improvement over full-finetuning TANL (Paolini et al., 2021). Fixed FISH mask (Sung et al., 2021) which showed sub-optimal performance in low-resource NER, also shows comparable performance in this scenario.

As we shift our focus towards more intricate tasks such as Joint Entity Relation extraction, FISH-DIP consistently exhibits superior performance compared to all baseline models. This task poses a significant challenge to all baselines operating in low-resource settings. As we dig deeper, we find that while some baselines may achieve a good entity-F1 score, they do so with a significant decrease in relation-F1 score. In contrast, FISH-DIP consistently matches or exceeds the performance of all other baselines, regardless of low-resource settings. Interestingly, the baseline model ASP (T5-large) exhibits comparable performance to FISH-DIP in higher resource settings, despite struggling in very low resource settings (e.g. 1% data). This can potentially be attributed to the task-specific parameterization of the ASP model. Although FISH-DIP may lead to even greater performance improvements when applied on ASP, it will compromise the generalization and multitasking capabilities fundamental to unified sequence labeling. Therefore, we only apply FISH-DIP for augmentation transformations as outlined in (Paolini et al., 2021).

Finally, in dialogue state tracking (Table 6) and semantic role labeling (Table 7), we observe a

| Method | FT Mask Sparsity | CoNLL'03 | | | OntoNotes | | |
|---|---|---|---|---|---|---|---|
| | | 1% | 5% | 10% | 1% | 5% | 10% |
| TANL | 100% | $81.31_{0.8}$ | $90.26_{0.4}$ | $90.68_{0.6}$ | $77.84_{0.9}$ | $83.78_{0.3}$ | $85.61_{0.3}$ |
| ASP (T5-large) | 100% | $49.70_{0.02}$ | $89.13_{0.3}$ | $89.24_{0.4}$ | - | - | - |
| Fixed FISH | 1% | $78.74_{3.8}$ | $86.98_{1.8}$ | $87.40_{0.4}$ | $75.00_{1.1}$ | $82.75_{0.2}$ | $83.30_{1.4}$ |
| Fixed FISH | 5% | $81.01_{1.0}$ | $87.08_{0.8}$ | $88.24_{0.6}$ | $76.21_{0.4}$ | $83.08_{0.4}$ | $84.88_{0.2}$ |
| FISH-DIP | 1% | $\mathbf{86.39_{0.4}}$ | $\mathbf{91.50_{0.6}}$ | $\mathbf{91.70_{0.6}}$ | $\mathbf{78.69_{0.4}}$ | $\mathbf{84.43_{0.3}}$ | $\mathbf{86.00_{0.4}}$ |

Table 3: Results for Named Entity Recognition. We report F1 scores in CoNLL'03 and OntoNotes datasets.

| Method | FT Mask Sparsity | FewRel | | | |
|---|---|---|---|---|---|
| | | 5-way 1-shot | 5-way 5-shot | 10-way 1-shot | 10-way 5-shot |
| BERT-EM | 100% | 88.9 | - | 82.8 | - |
| BERT-EM+MTB | 100% | 90.1 | - | 83.4 | - |
| BERT Pair | 100% | 85.7 | 89.5 | 76.8 | 81.8 |
| TANL | 100% | $87.2_{7.8}$ | $94.8_{3.6}$ | $83.6_{9.2}$ | $90.8_{5.2}$ |
| Fixed FISH | 1% | $\mathbf{96.0_{3.6}}$ | $96.8_{4.3}$ | $\underline{87.6_{5.0}}$ | $92.6_{4.3}$ |
| Fixed FISH | 5% | $\underline{94.8_{5.0}}$ | $98.3_{2.6}$ | $86.2_{6.0}$ | $\underline{94.2_{2.5}}$ |
| FISH-DIP | 1% | $92.8_{5.6}$ | $\mathbf{98.4_{2.6}}$ | $\mathbf{89.8_{5.6}}$ | $94.2_{2.1}$ |

Table 4: Results for Relation Extraction. We report F1 scores in FewRel dataset.

similar trend where FISH-DIP outperforms other baselines, particularly in extremely low-resource settings while matching or exceeding the performance of baselines in more resource-rich environments. In dialogue state tracking, we also evaluate the performance in more challenging 0.5% data, where TANL, Fixed FISH(1%), Fixed FISH (5%) and, FISH-DIP achieve joint accuracy scores of 34.17. 30.0, 30.9, and **35.87** respectively. These findings provide further evidence of the robustness of our FISH-DIP model irrespective of available resources.

## 4.2 Comparison against other PEFT approaches

Because of the expanding size of language models, over the past few years several techniques have been proposed for parameter-efficient fine-tuning (Houlsby et al., 2019; Karimi Mahabadi et al., 2021; Hu et al., 2021; Liu et al., 2022a; Li and Liang, 2021). Since FISH-DIP achieves notable performance improvements by leveraging a prudent sparse update procedure, we want to compare the performance benefit against other PEFT approaches. To this end, we compare the performance in Joint Entity-Relation CoNLL'04 dataset. We use Huggingface's PEFT library for LoRA (Hu et al., 2021) and Prefix Tuning (Li and Liang, 2021). We also also show the performance of Adapter (Houlsby et al., 2019), Compacter (Karimi Ma-

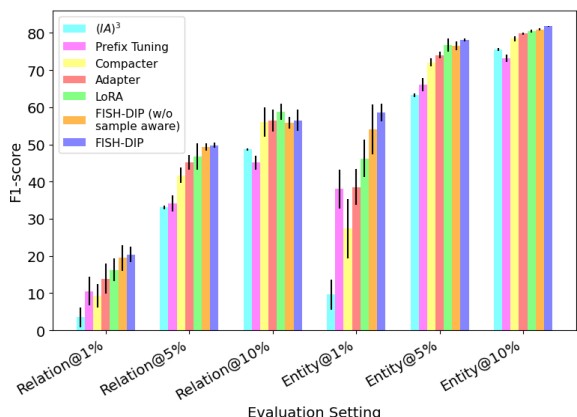

Figure 2: Performance comparison of FISH-DIP against different PEFT techniques in CoNLL'04 Joint Entity Relation dataset. FISH-DIP matches or exceeds all other approaches demonstrating its robustness in all low-resource settings.

habadi et al., 2021), and $(IA)^3$ (Liu et al., 2022a). Moreover, we show the performance of FISH-DIP without sample-awareness (parameter importance updates don't reflect regressing samples). All methodologies use identical hyperparameter configurations. The comparative outcomes are illustrated in Figure 2, indicating that FISH-DIP demonstrates either comparable or superior performance compared to other baseline approaches. This shows its robustness across various scenarios. Notably, under more demanding circumstances with limited data availability, other techniques experience performance degradation, whereas FISH-DIP consistently maintains optimal performance.

## 4.3 Comparison against In-Context Learning approaches with GPT-3.5-turbo

Recently, GPT-3.5-turbo gained attention for its high-quality text generation, supposed reasoning abilities, and impressive in-context learning (ICL) with minimal few-shot samples. We evaluate the performance of ICL with OpenAI's GPT-3.5-turbo

| Method | FT Mask Sparsity | CoNLL'04 | | | ACE'05 | | |
|---|---|---|---|---|---|---|---|
| | | 1% | 5% | 10% | 1% | 5% | 10% |
| | | Entity | | | | | |
| TANL | 100% | $46.05_{12.1}$ | $74.06_{2.1}$ | $79.11_{0.5}$ | $72.52_{2.1}$ | $81.35_{0.6}$ | $83.96_{0.8}$ |
| DygiePP | 100% | - | - | - | $59.64_{3.0}$ | $74.19_{1.0}$ | $77.82_{0.2}$ |
| ASP (T5-l) | 100% | $\mathbf{60.36_{0.2}}$ | $75.02_{0.5}$ | $\mathbf{82.34_{0.6}}$ | $61.55_{1.3}$ | $79.58_{0.7}$ | $\mathbf{84.68_{0.3}}$ |
| Fixed FISH | 1% | $37.08_{15.6}$ | $71.27_{3.0}$ | $77.27_{2.5}$ | $70.56_{1.9}$ | $79.57_{0.7}$ | $82.26_{0.7}$ |
| Fixed FISH | 5% | $51.37_{5.6}$ | $73.20_{1.2}$ | $79.24_{0.3}$ | $73.30_{2.0}$ | $80.00_{0.8}$ | $82.20_{0.3}$ |
| FISH-DIP | 1% | $56.88_{2.8}$ | $\mathbf{77.25_{1.2}}$ | $81.50_{0.9}$ | $\mathbf{75.08_{1.3}}$ | $\mathbf{82.76_{0.7}}$ | $84.64_{0.8}$ |
| | | Relation | | | | | |
| TANL | 100% | $14.13_{0.4}$ | $40.63_{3.6}$ | $54.37_{1.1}$ | $15.62_{2.5}$ | $36.79_{1.0}$ | $45.22_{1.1}$ |
| DygiePP | 100% | - | - | - | $7.00_{2.0}$ | $26.70_{1.7}$ | $45.74_{0.9}$ |
| ASP (T5-l) | 100% | $13.80_{0.2}$ | $40.93_{1.9}$ | $55.76_{0.9}$ | $6.40_{0.1}$ | $37.73_{1.9}$ | $\mathbf{47.85_{0.3}}$ |
| Fixed FISH | 1% | $12.29_{9.5}$ | $34.35_{4.5}$ | $46.32_{5.8}$ | $12.41_{1.9}$ | $29.07_{2.1}$ | $36.85_{1.2}$ |
| Fixed FISH | 5% | $19.40_{5.4}$ | $40.09_{2.5}$ | $53.88_{3.3}$ | $16.85_{3.5}$ | $33.84_{1.6}$ | $42.31_{2.0}$ |
| FISH-DIP | 1% | $\mathbf{21.63_{3.6}}$ | $\mathbf{49.20_{2.8}}$ | $\mathbf{57.08_{2.4}}$ | $\mathbf{19.01_{2.7}}$ | $\mathbf{39.85_{0.8}}$ | $47.15_{0.3}$ |

Table 5: Results for Joint Entity Relation Extraction. We report F1 scores in CoNLL'04 and ACE'05 datasets.

| Method | FT Mask Sparsity | MultiWoz 2.1 | |
|---|---|---|---|
| | | 1% | 5% |
| TRADE | 100% | 12.58 | 31.17 |
| SGPDST | 100% | 32.11 | 43.14 |
| DST - BART | 100% | 28.55 | 37.71 |
| DS2 - T5 | 100% | 33.76 | 44.20 |
| TANL | 100% | 34.96 | 47.37 |
| IC-DST (GPT-Neo-2.7B) | (ICL) | 16.70 | 26.90 |
| IC-DST (CodeGen-2.7B) | (ICL) | 20.72 | 29.62 |
| Fixed FISH | 1% | 34.17 | 41.70 |
| Fixed FISH | 5% | 37.50 | 45.70 |
| FISH-DIP | 1% | 42.33 | 47.24 |

Table 6: Results for Dialog State Tracking. We report Joint Accuracy in MultiWoz 2.1 dataset. Here (ICL) denotes in-context learning methods where finetuning is not involved.

| Method | FT Mask Sparsity | CoNLL'05 WSJ | | |
|---|---|---|---|---|
| | | 0.5% | 1% | 5% |
| TANL | 100% | $67.34_{1.5}$ | $74.50_{0.5}$ | $\mathbf{82.10_{0.3}}$ |
| Fixed FISH | 1% | $61.31_{0.8}$ | $65.60_{1.4}$ | $78.35_{1.3}$ |
| Fixed FISH | 5% | $66.53_{1.5}$ | $71.10_{0.5}$ | $79.56_{0.7}$ |
| FISH-DIP | 1% | $\mathbf{70.37_{0.3}}$ | $\mathbf{75.49_{0.4}}$ | $80.90_{0.2}$ |

| Method | FT Mask Sparsity | CoNLL'05 Brown | | |
|---|---|---|---|---|
| | | 0.5% | 1% | 5% |
| TANL | 100% | $58.71_{1.2}$ | $66.53_{0.8}$ | $\mathbf{73.69_{0.3}}$ |
| Fixed FISH | 1% | $51.00_{1.1}$ | $55.41_{0.9}$ | $68.36_{2.4}$ |
| Fixed FISH | 5% | $58.15_{2.7}$ | $62.86_{1.3}$ | $70.31_{1.2}$ |
| FISH-DIP | 1% | $\mathbf{62.10_{1.4}}$ | $\mathbf{68.07_{0.5}}$ | $72.42_{0.6}$ |

Table 7: Results for Semantic Role Labeling. We report F1 score in CoNLL'05 WSJ and Brown datasets.

API in CoNLL'04 and ACE2005 Joint Entity Relation datasets. The template used in prompting GPT-3.5-turbo is shown in Appendix C. We randomly select few-shot samples and demonstrate them as in-context pairs until reaching the token limit, leaving space only for the target input sentences. Finally, we give the test input sentence to generate output. The results are shown in Figure 3.

We find that the ICL performance varies significantly based on the chosen evaluation setting and dataset. In CoNLL'04, the performance of ICL in 1% data outperforms all other baselines in this setting. However, in 5% and 10% data, the numbers are suboptimal, demonstrating that it cannot take advantage of the extra data.In ACE'05 GPT-3.5-turbo shows very poor performance, especially in relation extraction. The complexity of ACE'05 with more complex entities and relation targets may have made it more difficult for the models to cap-

ture structural information for in-context learning.

## 4.4 Effect of Sample Aware Dynamic Sparse Tuning in Optimization

Figure 4 shows the trend of samplewise loss optimization for CoNLL'04 1% dataset. We find that across training steps, FISH-DIP focuses on optimizing samples with the highest losses because of selecting parameters associated with them. While doing that, parameters only relevant to the samples with lower losses are frozen, preventing irrelevant updates. In comparison, fixed sparsity mask results in samplewise irregular update trend, decreasing losses for some samples while increasing others. This illustrates the importance of dynamic parameter sparsity in few-sample optimization of PLMs.

## 4.5 Effect of Sparsity Percentage, $k$

For uniformity across all our experiments, we have opted for a stable parameter sparsity of $k = 1\%$ In Table 8, we analyze the influence of various

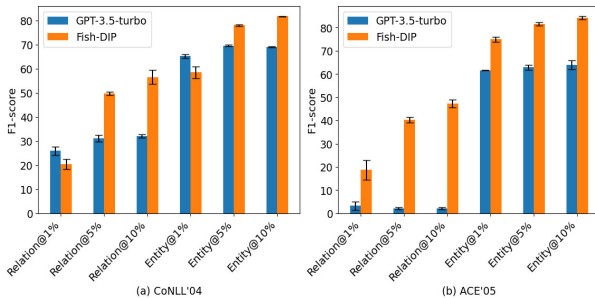
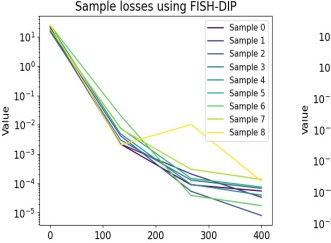
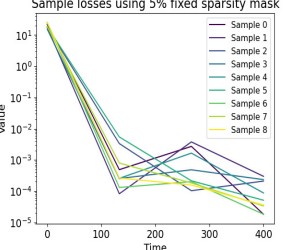

Figure 3: Comparison against in-context learning using GPT-3.5-turbo in different low resource setting in CoNLL'04 and ACE'05.

| | FT Mask | CoNLL'04 | | |
|---|---|---|---|---|
| | Sparsity | 1% | 5% | 10% |
| | 0.5% | $56.45_{1.1}$ | $76.37_{0.6}$ | $80.54_{0.7}$ |
| Entity | 1% | $\mathbf{58.52_{2.4}}$ | $\mathbf{78.05_{0.4}}$ | $\mathbf{81.37_{0.8}}$ |
| | 5% | $55.90_{3.9}$ | $76.67_{0.6}$ | $79.00_{0.1}$ |
| | 0.5% | $\mathbf{22.27_{3.2}}$ | $49.22_{1.4}$ | $56.03_{2.3}$ |
| Relation | 1% | $20.41_{2.1}$ | $\mathbf{49.78_{0.7}}$ | $\mathbf{57.03_{2.7}}$ |
| | 5% | $21.32_{5.1}$ | $46.41_{0.3}$ | $52.34_{1.5}$ |

Table 8: Effect of different sparsity choice (0.5%, 1%, 5%) in FISH-DIP on CoNLL'04

sparsity levels on the performance of FISH-DIP in different low-resource settings.

It is evident that the selection of parameter sparsity, $k$ has a potential impact on performance. While we have used 1% sparsity across all tests, tuning $k$ based on the available data can potentially yield improved results. This highlights the importance of carefully choosing the sparsity level in FISH-DIP. Hence, we recommend conducting experiments with different values of $k$ for achieving the optimal performance with FISH-DIP.

## 5 Related Work

**Unified Sequence Labeling** Paolini et al. (2021) noted out the issues in discriminative structured prediction tasks and proposed for an augmented Natural Lanugage format (Athiwaratkun et al., 2020) to complete all tasks in a sequence-to-sequence format. Following their success, (Liu et al., 2022b) has attempted to further improve the performance of these approaches through constrained generation which requires explicit constraints tailored towards each of the tasks effectively taking away the advantage of structured prediction unification. Unfortunately, in low-resource settings, the usefulness of these techniques have not been explored leaving them largely suboptimal in those settings.

Figure 4: Illustration of samplewise optimization of loss between FISH-DIP and 5% Fixed FISH mask sparse fine-tuning in CoNLL'04 (1% data). Across the timestamps, we see that FISH-DIP focuses on optimizing the samples having higher loss without compromising the learning of other samples. Compared to fixed sparse mask resulting in an overall smoother training trend.

**Low-Resource Sequence Labeling** Different learning approaches (Vinyals et al., 2016; Snell et al., 2017; Geng et al., 2019; Bao et al., 2019) have been leveraged in several NLP tasks to improve performance in few-shot settings. With the arrival of GPT-3 (Brown et al., 2020), in-context learning (Chen, 2022; Chen et al., 2023; Hu et al., 2022; Dong et al., 2022) started to become more commonplace for few-shot learning. However, in most previous works, few-shot/low-resource learning has been largely limited to sentence-level prediction, whereas sequence labeling tasks require better-structured prediction capabilities for improved performance.

Named Entity Recognition (NER) tasks are one of the first where different task-specific methods have been applied to improve few-shot NER performance (Yang and Katiyar, 2020; Das et al., 2021; Cui et al., 2021; Tong et al., 2021; Ma et al., 2022; Huang et al., 2022; Ming et al., 2022; Ji et al., 2022). In relation extraction, (Han et al., 2018) first introduced a large-scale benchmark for evaluation in N-way K-shot settings. This benchmark is later used by following works to improve few-shot performance in a wide range of domain-centered methods (Gao et al., 2019; Tong et al., 2021; Ma et al., 2022; Qu et al., 2020; Han et al., 2021). Other than that, (Hu et al., 2022; Wu et al., 2020; Lee et al., 2021; Shin et al., 2022) have applied in-context learning for dialogue state tracking, where they found that with in-context learning, gigantic language models can show noteworthy performance improvements over other models. These dedicated that few-shot techniques demonstrate improved performances but require extensive task-specific mod-

eling, limiting their applicability in a unified setting.

**Parameter Efficient Finetuning**  Different PEFT methods have been proposed for finetuning large language models efficiently. Small adapter layers that are added while keeping the rest of the network fixed was one of the first attempts at this efficient finetuning that proved to be quite useful (Houlsby et al., 2019). A follow-up approach (Karimi Mahabadi et al., 2021) leverages matrix decomposition and low-rank parameterization for adapter weights. Later (Sung et al., 2021) has introduced FISH methods for selection of parameters to be finetuned. On the other hand, recently (Hu et al., 2021) and (Liu et al., 2022a) introduced two other methods of PEFT where small number of new parameters are introduced for low-rank adaptation and scaled activation respectively. This was recently made even more efficient by Dettmers et al. (2023) significantly reducing memory usage. Another set of works have shown that it is possible to get good performance by injecting prompts into input and optimizing the related layers (Li and Liang, 2021). While these techniques have demonstrated their utility in facilitating efficient fine-tuning, they might not possess the capability to modify the output format necessary to accommodate the augmented language, especially the unified sequence labeling requirement in low-resource settings.

## 6   Conclusion

Unified sequence labeling poses significant difficulties, particularly when dealing with limited resources necessitating the adoption of a non-conventional augmented format. FISH-DIP tackles this challenge through dynamic sparse finetuning. It prioritizes underperforming instances and reduces the impact of well-learned samples, optimizing only a fraction of the parameters. Across five sequence labeling tasks, FISH-DIP demonstrated on-par or consistent improvement over all other baselines, particularly in more challenging extremely low-resource scenarios. For future research, exploring the potential of FISH-DIP in other low-resource/few-shot tasks (e.g. low-resource languages where external resources are extremely scarce) holds promise and warrants further investigation.

## Acknowledgement

We would like to thank Cisco Systems for their generous support through the Cisco Research Award for supporting this work. We would also like to thank Nan Zhang, Yusen Zhang, and Vipul Gupta for their valuable feedback and suggestions.

## Limitations

While we primarily focus on sequence labeling tasks, due to the broad applicability it would be interesting to explore the efficacy of FISH-DIP in other NLU tasks, particularly in low-resource or few-shot scenarios. Notably our approach is particularly effective in extreme low-resource settings. However, it is worth noting that as the scale of available data increases, most methods show comparable performance and room for improvement becomes narrower.

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

# A  Implementation Details

We'll primarily use Huggingface with PyTorch deep learning framework to implement and complete all our experiments. For the choice of PLM, we choose T5-large due to its better performance and moderate size (770 million). Our experiments are conducted on 8x NVIDIA A6000 48GB GPUs. For FISH-DIP, we choose sparsity $k = 1\%$, steps, $m = 100$, regressing examples, $n = 15$ as the hyperparameters. Consequently, *FISH* mask is recalculated after every $\approx \frac{m}{\lceil \frac{num\_samples}{batch\_size} \rceil}$ epochs. In low resource settings, given limited number of training samples, this recalibration should only introduce a minimal increment in training. While for simplicity, we fixed $m$ to 100, modulating $m$ dynamically, updating it more frequently during initial training phases and less so as the training progresses can be an interesting future direction towards further improvement. For fine-tuning hyperparameters (learning rate and epochs), we followed settings suggested in (Paolini et al., 2021).

For all other baselines, we used their respective implementations. For LoRA and Prefix Tuning, we used Huggingface PEFT with LoRA rank and prefix# in prefix tuning, chosen from 5, 10, 20, 30, and 40 (best performance).

# B  Task, Dataset, and Baseline Details

**Named Entity Recognition**  In named entity recognition (NER), given a sentence we need to extract all the relevant entities. In CoNLL'03 dataset, there are four target classes. and a training set size of 14k. On the other hand, OntoNotes have 18 classes with over 60k+ training samples. Table 3 shows the performance comparison in these two datasets respectively. We compare the performance against TANL full finetuning (Paolini et al., 2021), Fixed sparse mask finetuning (Sung et al., 2021). We also compare the performance against ASP (Liu et al., 2022b) which also models some structured prediction tasks into an autoregressive format, however, it also requires designing a task-specific constrained set of actions which restricts itself from being used in a unified scheme.

**Relation Extraction**  Given a sentence and head and tail entities, this task requires classifying the relation between them. In FewRel (Han et al., 2018) dataset, standard N-way K-shot settings is followed for few-shot performance evaluation. Following standard setting in this dataset, we meta-train the

model with 64 relations. During few-shot setting, we do the testing on 20 disjoint test relations. For these target relations, each of them have N-way K-shot support sets with which models are finetuned. Finally, inference is done on query set to calculate the Few-Shot performance. Table 4 compares the performance of FISH-DIP with TANL full finetuning (Paolini et al., 2021), fixed sparse FT (Sung et al., 2021), and several dedicated relation extraction methods for few shot relation extraction such as BERT-EM (+MTB) (Soares et al., 2019), BERT-PAIR (Gao et al., 2019)

**Joint Entity and Relation Extraction**  In this task, given a sentence, the target is to extract entities and a set of relations between pairs of entities. We evaluate the performance of FISH-DIP in Joint Entity and Relation extraction in CoNLL'04 and ACE2005 Joint ER dataset. While ACE2005 is moderately sized dataset having ∼7500 training samples, CoNLL'04 is a small dataset having ∼900 training samples. We show the performance comparison in CoNLL'04 and ACE2005 in Table 5. Like other experiments, we compare primarily against full finetuning TANL, fixed sparsity mask. Moreover, we also compare against the performance of popular method used in this domain - DygiePP (Wadden et al., 2019), and autoregressive method ASP (Liu et al., 2022b), both of which require task-specific modeling/predictions.

**Dialogue State Tracking**  In dialogue state tracking, given a history of dialogue turns between a user and an agent, who is assisting the user, we have to find the dialogue state, i.e. get the value for each slot from a predefined list. It has over 10k dialogue sets over seven distinct domains. We compare against TANL full fine-tuning (Paolini et al., 2021) and fixed sparsity finetuning (Sung et al., 2021). Moreover, we compared against dedicated DST methods (TRADE (Wu et al., 2020), SGPDST (Lee et al., 2021), DS2 - BART and DS2 - T5 (Shin et al., 2022)) for few-shot performance evaluation. For in-context learning-based method IC-DST (Hu et al., 2022) results are only reported with GPT-Neo 2.7B and CodeGen 2.7B. We don't include the results for Codex-DaVinci (reportedly 250 times larger than T5-large - 175 billion parameters) since OpenAI has discontinued support for it, which was also used for achieving state-of-the-art results in this task.

**Semantic Role Labeling** Given an input sentence and predicate, this task aims to predict a list of arguments and types to get the predicate-argument structure of a sentence. More specifically, we need to find all the arguments corresponding to the given predicate in the input sentence. Table 7 compares the performance of FISH-DIP with TANL full fine-tuning (Paolini et al., 2021) and fixed sparsity finetuning (Sung et al., 2021) in CoNLL'05 WSJ and Brown datasets.

# C In-Context Learning

For in-context learning with gpt-3.5-turbo (0301) for joint entity relation, randomly selected in-context demonstrations are drawn from the pool of samples within a low-resource environment, ensuring their adherence to the token limit of the model. In this regard, we use the following template:

# Instruction

Identify and mark all entities in the input sentence with square brackets [] and assign them from the following entity list: [list of entities]. If an entity has relation to another entity, indicate it with a vertical bar | and the name of the relation after the type. Relation type should be one from following list: [list of relations]. Some demonstrations are shown below:

# Demonstrations

Input: Tolkien wrote The Lord of the Rings.
Output: [ Tolkien | person ] wrote [ The Lord of the Rings | book | author = Tolkien ]
...
...
...

Input: (Test Sample)
Output: