# OpenReview forum: "Unified Low-Resource Sequence Labeling by Sample-Aware Dynamic Sparse Finetuning"
_EMNLP/2023/Conference — EMNLP 2023 Main_

### Official Review · Reviewer_XQLo · 2023-07-25

**Typos Grammar Style And Presentation Improvements:** Line 045 & Line 449 (both right column)
**Soundness:** 4

**Excitement:**

3: Ambivalent: It has merits (e.g., it reports state-of-the-art results, the idea is nice), but there are key weaknesses (e.g., it describes incremental work), and it can significantly benefit from another round of revision. However, I won't object to accepting it if my co-reviewers champion it.

**Paper Topic And Main Contributions:**

The paper studies a unified sequence labeling problem under the few-shot setting. The authors claim that most previous Parameter Efficient Fine-Tuning (PEFT) methods do not work very well in such settings, and they propose a new method FISH-DIP, which aims at selecting a small fraction of parameters in the pretrained language model (PLM) to update on the few-shot training sets, achieved via a sample-aware dynamic sparsity selection metric (Fisher information) with the help of feedback from highly regressing examples.

The authors conduct experiments on a series of sequence labeling tasks such as NER, RE, and SRL. The results demonstrate that the proposed method is able to outperform full fine-tuning and existing PEFT methods.

**Questions For The Authors:**

Please consider the **Reasons To Reject** questions.

**Reasons To Accept:**

* Novelty: Although the concrete techniques deployed in the framework (the unified sequence labeling augmentation format, the sparsity mask based on Fisher Information) are directly borrowed from previous work, their application to the sequence labeling setup seems new to me, which can be considered as the novelty of the work.
* Clarity: Most parts of the paper are clearly presented.
* Effectiveness: The FISH-DIP method is shown to outperform previous PEFT methods on a series of sequence labeling tasks under the few-shot setting. The empirical results seem comprehensive to me.

**Reasons To Reject:**

* Lack of a clear definition of "low-resource": The term "low-resource" is a broad concept and could mean either "few training data" or "few computational budget for updating model parameters". The methodological focus of this paper seems to be on the latter (i.e., selecting a subset of model parameters to update), while the former is more reflected in the experiment setting and seems to be a side-effect of the method. It would be better if the authors explicitly explain what they intend to mean by "low-resource" -- that would make the goal and focus of the paper more clear. Without the clarification, one would assume a lot more few-shot learning methods could be relevant baselines (e.g., self-training, data augmentation) that are currently missing from comparisons.
* Not sufficiently strong motivation: While the authors claim that previous PEFT methods do not perform satisfactorily on low-resource sequence labeling problems, there does not seem to be a very convincing argument as to why that could be the challenge, and why one should consider the sparsity parameter update methods in this paper. I'd like to see more conceptual/in-depth analyses/discussions in this regard (i.e., what makes FISH-DIP fundamentally different from other PEFT methods and how does it address their limitations?).

**Reproducibility:**

3: Could reproduce the results with some difficulty. The settings of parameters are underspecified or subjectively determined; the training/evaluation data are not widely available.

**Reviewer Confidence:**

4: Quite sure. I tried to check the important points carefully. It's unlikely, though conceivable, that I missed something that should affect my ratings.

---

> ### Author Rebuttal · Authors · 2023-08-29
>
> **W1:** We thank you for bringing up the matter of the "Low Resource" terminology. Our primary objective within this paper revolves around enhancing sequence labeling performance within the constraints of **limited data scenarios** (similar to UIE [1]), all the while adhering to a unified and versatile approach to sequence labeling tasks. While we indeed draw motivation from a related PEFT approach, namely the FISHer information mask [2], this choice serves to bolster the overarching generalizability of our methodology across an extensive array of sequence labeling tasks. Notably, our proposed method does not need any supplementary pretraining with external resources, thereby rendering it adaptable to languages with limited resources. This characteristic significantly amplifies the "generalizability" aspect within the scope of low-resource sequence labeling.
>
> In summary, our methodological selection is driven by a commitment to a universally applicable strategy for enhancing sequence labeling performance. Regarding baselines, we believe we have compared against the most popular state-of-the-art techniques commonly used on each dataset, demonstrating consistent performance improvement across the board.
>
> **W2:** Thank you for your comment. Due to our adherence to a unified approach to sequence labeling, it is imperative to adapt base PLMs to an augmented natural language format. Most contemporary PEFT approaches ((IA)^3, Adapter, Compacter, etc.) introduce a small amount of trainable new parameters during finetuning while keeping the rest of the network fixed during the finetuning process. However, in the realm of generative sequence labeling, it is very difficult to accommodate the augmented linguistic structure by training only these newly added parameters due to the scarcity of training data, as empirically demonstrated in Figure 2 and the newly added results of ACE'05 Joint ER task in 1% data in R2 - W2/Q2 (added here for ease of referencing)
>
> | Method   | Entity F1 | Relation F1 |
> |----------|-----------|-------------|
> | Adapter  | 72.96     | 14.08       |
> | LoRA     | 71.87     | 13.25       |
> | (IA)^3   | 63.54     | 6.7         |
> | FISH-DIP | 75.11     | 19.26       |
>
> To address this inherent challenge, we propose sparsity dynamism suited for few-sample sequence labeling scenarios based on the foundation of the FISHer information mask [2], an approach that selects parameters for sparse fine-tuning. This strategy ensures that our method retains its effectiveness even within the constraints of extremely limited training data.

---

### Official Review · Reviewer_C1eS · 2023-08-03

**Soundness:** 3

**Excitement:**

3: Ambivalent: It has merits (e.g., it reports state-of-the-art results, the idea is nice), but there are key weaknesses (e.g., it describes incremental work), and it can significantly benefit from another round of revision. However, I won't object to accepting it if my co-reviewers champion it.

**Paper Topic And Main Contributions:**

This paper proposes a FISH-DIP method for better fine-tuning of PLMs on sequence labeling tasks. The FISH-DIP strategy is a kind of parameter-efficient finetuning method and is based on Fisher Information sparsity training while realizing dynamic sparse training with parameter selection during training. The authors have done many experiments to show the effectiveness of FISH-DIP on seven datasets. They also have done some analytical experiments to further illustrate the method.

**Questions For The Authors:**

1. Please clear up my confusion in *Reasons To Reject 1.

2. Why is FISH-DIP better than other parameter-efficient tuning methods (such as LoRA) in low-resource settings? I didn't figure out what advantages of FISH-DIP can definitely lead to better results.

**Reasons To Accept:**

1. The idea of sample-aware parameter selection is interesting. And the experiments show it is also useful.

2. The authors have done extensive experiments on 7 datasets. The results seem to be solid.

**Reasons To Reject:**

1. The motivation of this paper is quite confusing and inconsistent. In fact, I didn't figure out what's the relationship between "Sequence-labeling tasks", "low-resource settings" and the FISH-DIP method. In my opinion, the method can be used in any other tasks that require fine-tuning from a PLM. Also, there's no point to show that FISH-DIP is definitely better for low-resource settings. At least the introduction part didn't persuade me.

2. I think there should include more baselines: (1) The idea of the sample-based parameter selection of FISH-DIP is similar to re-weighting the training loss (enlarging the weight of the loss of the more difficult samples). Would this be a reason that FISH-DIP is better than other PEFT methods? I think this needs further discussion. (2) As said in 1, I still wonder why the authors only chose the sequence-labeling tasks since I don't think the claimed problem is unique in sequence labeling. But if the sequence labeling task is chosen, then more baselines such as UIE [1] (that also solve the problem in low-resource settings) need to be compared. (3) The FISH-DIP is still a PEFT method, so why  not compare the PEFT methods on all datasets? I still can't figure out why FISH-DIP is definitely better than other PEFT method in low-resource settings, and the results in Figure 2 is not so promising.

[1] Lu, Yaojie, et al. "Unified structure generation for universal information extraction." arXiv preprint arXiv:2203.12277 (2022).

**Reproducibility:**

3: Could reproduce the results with some difficulty. The settings of parameters are underspecified or subjectively determined; the training/evaluation data are not widely available.

**Reviewer Confidence:**

4: Quite sure. I tried to check the important points carefully. It's unlikely, though conceivable, that I missed something that should affect my ratings.

---

> ### Author Rebuttal · Authors · 2023-08-29
>
> **W1 / Q1:** We sincerely thank you for your feedback. The principal objective of our paper is to present a generalizable, robust, unified framework aimed at enhancing performance across various sequence labeling tasks in low-resource scenarios. This can be highly useful in a broad range of downstream applications from preserving indigenous languages to enhancing knowledge transfer across different domains and crisis communication. Despite its significance, this task remains largely unexplored within the research community. The inherent scarcity of annotated data in this setting requires a versatile approach to low-resource sequence labeling, which can also be extremely useful for a wide range of downstream applications.
>
> Given the highly varied natures of different sequence labeling tasks, crafting a universally adaptable solution presents inherent challenges. Our introduced methodology, FISH-DIP, is tailored to address this very issue. By being ingrained at the PLM fine-tuning level, FISH-DIP demonstrates commendable adaptability, seamlessly integrating with simple sequence-to-sequence structured prediction methods, and without needing any supplementary data or structural modifications. We posit this flexibility as a primary feature of our proposition. Its adaptability not only enhances performance in low-resource scenarios but also does so across a wide range of structured prediction tasks that are very different from each other.
>
> **W2 / Q2:** Thanks for the insights regarding the motivation and applicable baselines. Our replies are given sequentially for each of them:
>
> (1) **Difference between FISH-DIP and weight amplification:** Emphasizing challenging samples during training is certainly a motivation for our approach. While enlarging the weight of more difficult samples is a possible approach to attain that effect, it risks distorting the data distribution and promoting overfitting to the limited available samples. Conversely, we do not alter any weight to the training losses to achieve maximum learning from the given samples. Instead, we only allow the training of the parameters pertinent to the difficult samples while keeping the rest of the parameters frozen. In this way, we improve low-resource performance while maximizing learning from the small number of samples.
>
> (2) **Motivation for Sequence Labeling and Related Methods:** As highlighted in W1/Q1, sequence labeling encompasses a diverse array of problems that hold substantial implications for various downstream applications, especially in resource-constrained contexts—a facet that remains inadequately explored within the research landscape. Consequently, we apply and evaluate our method in sequence labeling tasks that are quite varied from each other, to show the effectiveness of FISH-DIP in different scenarios.
>
> Turning to baseline methodologies, UIE [1] emerges as particularly relevant within the realm of sequence labeling. Methodologically, it expresses sequence labeling in a structured format and pretrains a T5 model using large-scale additional resources, resulting in a modified pretrained model called UIE. This model can then be fine-tuned for different Information extraction tasks.
>
> In our paper, across all baselines, we have kept the pretrained language model fixed to T5-large (please refer to R1 W1), to focus on the sequence labeling methodology itself rather than the pretrained language model. Furthermore, pretraining a language model with large-scale additional resources might not be possible in lower-resource languages, which significantly limits its usefulness and defeats the purpose of generalization (multilingual models such as mT5 can be a good PLM in these scenarios to use FISH-DIP).
>
> Nevertheless, we summarize the performance of FISH-DIP and UIE in the CoNLL’03 NER task in the table below, the only task-dataset that is common to both the papers. We also add the performance with T5-base applied with FISH-DIP for better comparison. The table demonstrates that without any large-scale pretraining structured schema of UIE may offer suboptimal performance. Still, FISH-DIP offers commendable performance regardless of the PLMs used, proving its robustness in sequence labeling. We will include this comparison in the final version of our manuscript as well.
>
> | Method                | Large Scale Pretraining | F1-Score @1% Data | F1-Score @5% Data | F1-Score @10% Data |
> |-----------------------|-------------------------|-------------------|------------------|-------------------|
> | UIE (T5-v1.1-base)    | No                      | 75.74             | 85.71            | 87.70             |
> | UIE (base)    | Yes                     | 82.84             | 88.34            | 89.63             |
> | FISH-DIP (T5-base)    | No                      | 82.38             | 91.24            | 91.83             |
> | FISH-DIP (T5-large)   | No                      | 85.83             | 91.50            | 91.70             |
>
> (3) **Comparison with other PEFT approaches in low-resource setttings** We acknowledge that it would have been ideal to compare FISH-DIP against all PEFT approaches in all settings. However, given the large number of experiments (including each experiment done at least three times to calculate variance), it is quite difficult to cover all combinations in all settings. For rebuttal purposes, we are including some additional experiment results done on ACE’05 Joint ER task in the 1% data setting for popular PEFT approaches - Adapter [3], LoRA [4], and (IA)^3 [5]. We will add more comparisons in our final version:
>
> | Method   | Entity F1 | Relation F1 |
> |----------|-----------|-------------|
> | Adapter  | 72.96     | 14.08       |
> | LoRA     | 71.87     | 13.25       |
> | (IA)^3   | 63.54     | 6.7         |
> | FISH-DIP | 75.11     | 19.26       |
>
> As for the reason for the effectiveness of FISH-DIP compared to other PEFT approaches, learning a unified sequence labeling approach requires adapting base PLMs to an augmented natural language structure. Contemporary PEFT techniques introduce new trainable parameters during fine-tuning. However, in the realm of generative sequence labeling, it is very difficult to accommodate the augmented linguistic structure by training only these newly added parameters due to the scarcity of training data, as empirically demonstrated in Figure 2. To address this, we propose dynamic parameter selection for sparse finetuning, building on the FISHer information mask [2], enabling our method to maintain effectiveness under limited data availability.
>
> Finally, in Figure 2 of the original manuscript, we present the performance of the CoNLL'04 Joint Entity Relation Extraction task. Within each data setting (1%/5%/10%), we delve into both entity and relation extraction performance. A careful observation of this figure reveals a consistent trend: FISH-DIP consistently outperforms all other PEFT approaches. Particularly noteworthy is its performance in the most challenging scenario, where only 1% of the data is available. In this case, FISH-DIP maintains a high level of performance, whereas every other method experiences a significant performance decline in both entity and relation extraction tasks.
>
> This observation serves as a compelling illustration of how FISH-DIP is better suited for low-resource sequence labeling tasks, as depicted in Figure 2 of the paper.
>
>
> [1] Lu, Yaojie, et al. "Unified structure generation for universal information extraction." arXiv preprint arXiv:2203.12277 (2022).
>
> [2] Sung, Yi-Lin, Varun Nair, and Colin A. Raffel. "Training neural networks with fixed sparse masks." Advances in Neural Information Processing Systems 34 (2021): 24193-24205.
>
> [3] Houlsby, Neil, et al. "Parameter-efficient transfer learning for NLP." International Conference on Machine Learning. PMLR, 2019.
>
> [4] Hu, Edward J., et al. "LoRA: Low-rank adaptation of large language models." arXiv preprint arXiv:2106.09685 (2021).
>
> [5] Liu, Haokun, et al. "Few-shot parameter-efficient fine-tuning is better and cheaper than in-context learning." Advances in Neural Information Processing Systems 35 (2022): 1950-1965.

---

### Official Review · Reviewer_YNan · 2023-08-11

**Soundness:** 4

**Excitement:**

4: Strong: This paper deepens the understanding of some phenomenon or lowers the barriers to an existing research direction.

**Paper Topic And Main Contributions:**

This paper proposes FISH-DIP, an improvement to the parameter-efficient fine-tuning that introduces Dynamic Importance of Parameters (DIP) to the Fisher information method for sparse fine-tuning. The method only uses top k% of samples that produce the highest loss (i.e. most regressing samples) to calculate an empirical approximation to the Fisher information for parameter importance, and produces a FISH mask for sparse parameter updates to the most important parameters. The approach is evaluated in a low-resources setting within a unified sequence labeling framework, taking the T5 large model as baseline, on a variety of information extraction datasets. In the low-resource setting, experimental results demonstrate FISH-DIP's competitive or superior performance to full fine-tuning baselines, fixed importance-based parameter update methods, or to in-context learning LLMs.

**Questions For The Authors:**

A. How much computational overhead does FISH-DIP introduce compared to the baselines? Does it significantly slow down the training? What are the effects of varying the hyper-parameter m (update steps)?

**Reasons To Accept:**

This paper proposes a simple and effective extension to Fisher-information-based fine-tuning methods with a fixed sparsity mask. The benefit of this extension is convincingly demonstrated within the low-resource sequence2sequence setting on multiple standard information extraction benchmarks. The positive per-sample effect of dynamic sparse tuning vs fixed sparse tuning is illustrated in by the smooth sample-wise optimization plot in Figure 4.

**Reasons To Reject:**

- It is inferred but not spelled out that the baseline (T5 large) is the same for FISH-DIP vs Fixed FISH 1% and Fixed FISH 5%. This should be spelled out more clearly in the experiments section, not the appendix.
- Time and memory efficiency of dynamically re-computing the parameter importance and sparsity matrices is not discussed.
- How does this method extend to truly low-resource settings, such as information extraction for low-resource languages?

**Reproducibility:**

4: Could mostly reproduce the results, but there may be some variation because of sample variance or minor variations in their interpretation of the protocol or method.

**Reviewer Confidence:**

3: Pretty sure, but there's a chance I missed something. Although I have a good feel for this area in general, I did not carefully check the paper's details, e.g., the math, experimental design, or novelty.

---

> ### Author Rebuttal · Authors · 2023-08-29
>
> We greatly appreciate the comprehensive review and insightful comments provided. Please find our responses to your comments below:
>
> **W1:** We acknowledge the oversight regarding the clear specification of the baseline model inside the original draft. In our revised manuscript, we will explicitly state that all models utilize *T5-large* as the baseline. This should enhance clarity and eliminate any potential ambiguities.
>
> **W2/Q1:** We are grateful for highlighting this aspect. Since our paper mainly revolves around very low resource sequence labeling in a unified context, its emphasis is on the efficacy of our approach relative to other methods. That being said, on the computational front, the efficiency of *FISH-DIP* can, at its best, match that of the original *FISH* mask, but this is contingent on the number of necessary training steps. The updating frequency of the *FISH* mask, set at $m$ steps, implies that the mask is recalculated after every $ \approx m/ \lceil {num\\_samples}/ {batch\\_size} \rceil $ epochs. Given the limited training samples in our low-resource context, this recalibration should introduce only a minimal increment in training duration.
>
> Furthermore, regarding the hyperparameter $m$, we've standardized it to 100 training steps for simplicity. Nevertheless, an intriguing avenue for future work could be to modulate $m$ dynamically, updating it more frequently during initial training phases and less so as training progresses.
>
> To reiterate, our central focus is on the performance outcomes of unified sequence labeling. Consequently, a deep dive into these computational intricacies somewhat extends beyond our paper's intended scope.
>
> **W3:** This is a very good point and we concur with your observation. The advantage of our proposed technique is its independence from additional datasets/resources and its reliance on a generalizable strategy to augment performance. This characteristic makes it especially potent for low-resource languages where external resources are extremely limited. As a natural extension, its application with multilingual-T5 variants to further enhance performance in similar low-resource language tasks might be very interesting.

---

### Meta-Review · Area_Chair_vHMA · 2023-09-20

**Recommendation:** 4

**Metareview:**

This paper proposes FISH-DIP to use fisher information to highlight those task-specific parameters and then tune these highlighted parameters. In fact, using fisher information to highlight important parameters is widely used in continual learning, which is not a novel method. Considering that the paper is well organized and clearly presented, it is appropriate for this paper to be accepted to the main conference or findings

---

### Decision · Program_Chairs · 2023-10-07

**Decision:**

Accept-Main

**Comment:**

This paper proposes FISH-DIP to use fisher information to highlight those task-specific parameters and then tune these highlighted parameters. In fact, using fisher information to highlight important parameters is widely used in continual learning, which is not a novel method. Considering that the paper is well organized and clearly presented, it is appropriate for this paper to be accepted to the main conference or findings